# Learning Tractable Probabilistic Models from Inconsistent Local Estimates

**Shasha Jin, Vasundhara Komaragiri, Tahrima Rahman, and Vibhav Gogate**
The University of Texas at Dallas

## Abstract

Tractable probabilistic models such as *cutset networks* which admit exact linear time posterior marginal inference are often preferred in practice over intractable models such as Bayesian and Markov networks. This is because although tractable models, when learned from data, are slightly inferior to the intractable ones in terms of goodness-of-fit measures such as log-likelihood, they do not use approximate inference at prediction time and as a result exhibit superior predictive performance. In this paper, we consider the problem of improving a tractable model using a large number of local probability estimates, each defined over a small subset of variables that are either available from experts or via an external process. Given a model learned from fully-observed, but small amount of possibly noisy data, the key idea in our approach is to update the parameters of the model via a gradient descent procedure that seeks to minimize a convex combination of two quantities: one that enforces closeness via KL divergence to the local estimates and another that enforces closeness to the given model. We show that although the gradients are NP-hard to compute on arbitrary graphical models, they can be efficiently computed over tractable models. We show via experiments that our approach yields tractable models that are significantly superior to the ones learned from small amount of possibly noisy data, even when the local estimates are inconsistent.

## 1 Introduction

A major issue with probabilistic graphical models (PGMs) [3, 8] such as Bayesian and Markov networks is that in most real-world domains, exact inference is computationally infeasible. As a result, approximate inference algorithms are often used in practice, but these algorithms can be inaccurate and may exhibit high variability. Thus, even if a PGM performs well on *goodness-of-fit* measures such as test set log-likelihood, its predictive performance, because of the use of approximate inference methods, can be quite poor.

To combat this problem, the field of tractable probabilistic models (TPMs) or probabilistic circuits (PCs) [9, 11, 20] seeks to learn probabilistic models that admit polynomial time exact inference algorithms. TPMs are more biased or less expressive than general probabilistic models (because they enforce tractability) and as a result may yield a slightly inferior (model) fit. But, the hope is that because no approximations are used at prediction time the slight drop in goodness-of-fit measures is offset by superior predictive performance. This hope is often realized in practice, and demonstrated in numerous empirical studies in literature (for example, see [18, 20]).

To date, algorithms for learning the structure and parameters of tractable probabilistic models such as cutset networks [17], sum-product networks [14] and probabilistic sentential decision diagrams [9] assume access to either full data which has no missing values or almost full data in which only a few variables have missing values. Often in the real world, however, the following scenario is quite common. The learning algorithm has access to a small amount of full, possibly noisy data and a large

number of local marginal probability estimates that are derived from some combination of local data, imperfect domain knowledge and local, independent predictive models. For instance:

- Due to privacy concerns in application domains such as social networks [7, 22], only limited global data is available. But local statistics, such as information about a person's contacts/connections can be retrieved easily.
- In lazy learning of generative models [18, 26, 27], we derive sufficient statistics needed for inducing a probabilistic model at test time (when a query is made) from various sources such as local classifiers for each statistic or via a query made to a large database.
- In active learning [23], the learning algorithm interactively solicits a user for labels or in general local marginal probability distributions for certain variables that the user is an expert at given observations.

A hallmark of all the scenarios just outlined is that the local marginal estimates are often *inconsistent*, namely, there may not exist a joint probability distribution that is consistent with the local estimates. For example, in the active learning case, consider two pairwise marginal estimates $P_A(x_1, x_2)$ and $P_B(x_2, x_3)$ over three random variables $X_1, X_2, X_3$ derived by querying two users $A$ and $B$ respectively. A joint probability distribution $P(x_1, x_2, x_3)$ over the three variables that is consistent with the two estimates exists iff $P_A(x_2) = P_B(x_2)$ where $P_A(x_2) = \sum_{x_1} P_A(x_1, x_2)$ and $P_B(x_2) = \sum_{x_3} P_B(x_2, x_3)$. Unfortunately, because of factors such as precision errors and user bias, $P_A(x_2)$ will not be equal to $P_B(x_2)$.

In this paper, we focus on the following learning problem over TPMs. We assume that the learner has access to: (1) a TPM learned from a small dataset having few or no missing variables, and (2) a set containing *local estimates* where each estimate is a marginal probability distribution over a small subset of variables given observations.

**Contributions.** We propose to express the above learning task as the following minimization problem: update the parameters of the given tractable model such that a linear combination of the following two KL distances is minimized: (1) the distance between the distribution represented by the original parameters and the one represented by the updated parameters; and (2) the sum of the distances between the given local, possibly inconsistent estimates and the ones computed using the updated distribution. We derive a gradient-based method for solving this optimization task. Since the gradients require computation of marginal distributions over subsets of variables, they are NP-hard in general on arbitrary probabilistic models but can be computed efficiently on tractable models [24]. This shows the virtue of using tractable models in our learning settings.

We performed a controlled empirical evaluation of our proposed method using 20 popular datasets that have been used in numerous studies on tractable models [10]. Our results show that our approach that leverages local estimates yields significant improvements in both generative and predictive performance over the original model learned from small amount of training data, even when the local estimates are inconsistent. Moreover, since the optimization problem is smooth, our procedure is guaranteed to reach a local optima under mild conditions.

## 2  Related Work

Vomlel [25] studied the problem of integrating probabilistic knowledge bases where a joint probability distribution is constructed from low-dimensional probability distributions (local estimates). Vomlel used a classic optimization method called iterative proportional fitting procedure (IPFP) [6] and proposed a variant called the generalized expectation maximization algorithm (GEMA) for solving this problem. Vomlel provided convergence proofs for these methods; showing that IPFP converges when the local estimates are consistent and GEMA converges even if the local estimates are inconsistent. Unfortunately, Vomlel's approach has high computational complexity (is exponential in the treewidth of the graph defined over the local estimates) and is not practical. The method proposed in this paper does not have this limitation. Peng and Ding [13] proposed two polynomial time approximations for IPFP and applied them to Bayesian networks. However, their preliminary experimental study demonstrates that the error due to their approximations is quite high and convergence is not guaranteed if the local estimates are inconsistent. In contrast, our proposed method makes very few approximations and leverages tractable inference to yield a practical scheme.

## 3   Notation and Background

We use the following notation. Bold upper-case letters (e.g., $\boldsymbol{X}$, $\boldsymbol{Y}$, $\boldsymbol{U}$, etc.) are used to denote sets of discrete random variables while bold lower-case letters (e.g., $\boldsymbol{x}$, $\boldsymbol{y}$, etc.) denote an assignment of values to all variables in the corresponding set denoted by bold upper case letters (thus $\boldsymbol{x}$ and $\boldsymbol{y}$ denote an assignment to all variables in the set $\boldsymbol{X}$ and $\boldsymbol{Y}$ respectively). We use upper-case letters (e.g., $X, U, V$, etc.) to denote the variables. For simplicity of exposition, we assume that all variables are binary taking values from the set $\{0, 1\}$. A lower case letter (e.g., $x, y$, etc.) denotes an assignment of a value to the corresponding variable denoted by the upper case letter.

For simplicity of exposition, we present our algorithm and experimental results on a specific type of tractable model called cutset networks [17] which combines Bayesian networks and AND/OR graphs [5], noting that the algorithm in this paper can be easily extended to other tractable models such as sum-product networks [14], thin junction trees [1], and arithmetic circuits [2].

### 3.1   Bayesian Networks

Bayesian networks (BNs) [3, 12] use directed acyclic graphs (DAGs) and conditional probability tables (CPTs) to compactly represent joint probability distributions over a set of random variables. Each node in the DAG denotes a random variable and is associated with a conditional probability distribution of the corresponding variable given its parents in the DAG. Formally, a Bayesian network is a triple $\langle \boldsymbol{X}, G, F \rangle$ where $\boldsymbol{X} = \{X_1, \dots, X_n\}$ is a set of random variables; $F = \{F_1, \dots, F_n\}$ is a set of conditional probability tables (CPTs); and $G(V, E)$ is a directed acyclic graph such that each vertex $V_i \in V$ is associated with the variable $X_i \in \boldsymbol{X}$ and $E$ is the set of directed edges. Each node $V_i$ is associated with the CPT $F_i = P_i(X_i | \boldsymbol{U}_i)$ that represents the conditional probability distribution of the variable $X_i$ given its parents $\boldsymbol{U}_i \subseteq \boldsymbol{X} \setminus X_i$.

Let $\boldsymbol{x} = (x_1, \dots, x_n)$ be an assignment of values to all variables in the set $\boldsymbol{X}$ and let $\boldsymbol{u}_i \sim \boldsymbol{x}$ denote the value assignment to all variables in the set $\boldsymbol{U}_i$ that is compatible with $\boldsymbol{x}$, namely $\boldsymbol{u}_i$ is the projection of $\boldsymbol{x}$ on $\boldsymbol{U}_i$. We will parameterize the Bayesian network using a set of parameters $\Theta$ where each $\theta_{x_i \boldsymbol{u}_i} \in \Theta$ is equal to the conditional probability $P(X_i = 1 | \boldsymbol{U}_i = \boldsymbol{u}_i)$. Note that since $P(X_i = 1 | \boldsymbol{U}_i = \boldsymbol{u}_i) + P(X_i = 0 | \boldsymbol{U}_i = \boldsymbol{u}_i) = 1$, we do not need to have a separate parameter for $P(X_i = 0 | \boldsymbol{U}_i = \boldsymbol{u}_i)$. Under this parameterization, the probability distribution represented by the Bayesian network can be written as:

$$P(\boldsymbol{x}) = \prod_{\substack{i=1 \\ \boldsymbol{u}_i \sim \boldsymbol{x}}}^{n} (\theta_{x_i \boldsymbol{u}_i})^{x_i} (1 - \theta_{x_i \boldsymbol{u}_i})^{1 - x_i} \tag{1}$$

The two main reasoning tasks over Bayesian networks are most-probable explanation (MPE) and posterior marginal inference (MAR). The former seeks to find an assignment of values to all variables given evidence such that the probability is maximized. Formally, let $\boldsymbol{E} \subset \boldsymbol{X}$ be a set of evidence variables, let $\boldsymbol{Z} = \boldsymbol{X} \setminus \boldsymbol{E}$, and $\boldsymbol{e}$ be an assignment to all variables in $\boldsymbol{E}$, then the MPE task is given by $\arg\max_{\boldsymbol{z}} P(\boldsymbol{z}, \boldsymbol{e})$. The MAR task seeks to find the marginal probability distribution over each non-evidence variable given $\boldsymbol{e}$, namely compute $P(z_j | \boldsymbol{e})$ for each variable $Z_j \in \boldsymbol{Z}$. Both tasks are known to be NP-hard in general but can be solved efficiently in practice on Bayesian networks having *small treewidth* using bucket (or variable) elimination and propagation algorithms [4]. Specifically, given a Bayesian network having treewidth $w$, the time and space complexity of these exact inference algorithms is bounded by $O(n2^{w+1})$ and $O(n2^w)$ respectively.

### 3.2   Cutset Networks

Cutset networks (CNs) [15, 17] are tractable probabilistic models that combine two well-known classes of tractable models: AND/OR graphs [5] and tree Bayesian networks. Formally, a CN defined over a set of variables $\boldsymbol{X}$ ($\boldsymbol{X}$ may include latent variables) is defined recursively using the following three conditions: (1) A tree Bayesian network over $\boldsymbol{X}$ is a CN; (2) An OR node labeled by a variable $X_i \in \boldsymbol{X}$ such that $|\boldsymbol{X}| > 1$ with two child CNs, each defined over the set $\boldsymbol{X} \setminus \{X_i\}$ is a CN. We follow the convention that the left child of the OR node labeled by $X_i$ represents conditioning over $X_i = 0$ and the right child represents conditioning over $X_i = 1$. The edges from the OR node to its child nodes are labeled with probability values in $\mathbb{R}^+$ such that they sum to 1; and (3) Let $(\boldsymbol{X}_1, \boldsymbol{X}_2)$

be a partition of $\boldsymbol{X}$ such that $|\boldsymbol{X}| > 1$. Then, an AND node with two child CNs, one defined over $\boldsymbol{X}_1$ and the second defined over $\boldsymbol{X}_2$ is a CN.

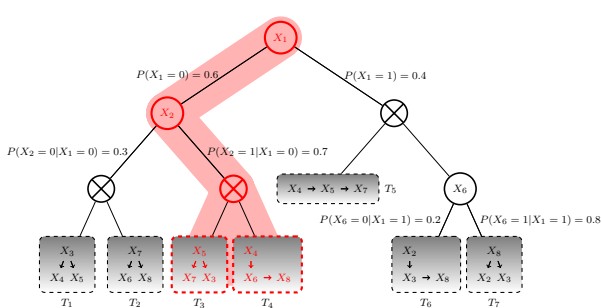

Figure 1: A cutset network defined over eight binary variables $\{X_1, \ldots, X_8\}$. OR nodes are labeled with variables and denoted by circles, AND nodes by cross mark in circles and leaf nodes (tree Bayesian networks $\mathcal{T}_1, \ldots, \mathcal{T}_7$) by shadowed dotted rectangles. Left and right edges emanating from an OR node correspond to an assignment of 0 and 1 respectively to the corresponding variable and are labeled with conditional probabilities. For instance, the nodes and edges highlighted in red show the sub-tree consistent with the assignment $\{X_1 = 0, X_2 = 1\}$.

Fig. 1 shows a CN defined over eight variables. In general, a full assignment $\boldsymbol{x}$ yields a rooted sub-graph in a CN.

Following our notation for Bayesian networks, let $\Theta$ denote the set of parameters of the cutset network, such that $\theta_{x_i \boldsymbol{u}_i} \in \Theta$ is equal to the conditional probability $P(X_i = 1 | \boldsymbol{U}_i = \boldsymbol{u}_i)$. Given an assignment $\boldsymbol{x}$, when $X_i$ is an OR node, $\boldsymbol{u}_i$ denotes the assignment along the path from the root of the CN to $X_i$. Alternatively, when $X_i$ belongs to a tree Bayesian network in the CN, $\boldsymbol{u}_i$ denotes the assignment formed by composing the assignment along the path from the root of the CN to the tree Bayesian network with the assignment to the parent of $X_i$ (if $X_i$ has a parent in the network). Given this parameterization, the probability distribution associated with the cutset network is given by

$$P(\boldsymbol{x}) = \prod_{\substack{i=1 \\ \boldsymbol{u}_i \sim \boldsymbol{x}}}^{n} (\theta_{x_i \boldsymbol{u}_i})^{x_i} (1 - \theta_{x_i \boldsymbol{u}_i})^{1-x_i} \tag{2}$$

A distinguishing feature of CNs is that when they have no latent variables, both MAR and MPE inference can be performed over them in time that scales linearly with the size of the network. This can be accomplished by converting the CN in linear time to an AND/OR graph [5] or an arithmetic circuit [2] and performing two-passes over these circuits. When latent variables are present, CNs admit linear time MAR inference only while MPE inference is intractable in general.

## 4 Our Approach

In this section, we define the optimization problem for learning the parameters of a cutset network from local estimates and present a gradient-based algorithm for solving it.

### 4.1 The Learning Problem

The high-level goal of this paper is to improve the prediction quality and model fit of an existing model that is induced using global information or full data alone by leveraging *local information*. The latter is often available in practice from various sources such as prior knowledge, expert domain knowledge, external sensors or processes and (local) data over a small subset of features. A distinguishing property of local information is that it is typically available in large quantities and thus more robust as compared to global information/full data. However, because of various issues such as user bias (e.g., when information is obtained from different experts), locality and privacy constraints, local information is often noisy and inconsistent. Therefore, our specific goal is to learn cutset networks such that they optimally combine both local and global information, regularizing the two appropriately based on background knowledge.

We begin by describing required notation and simplifying assumptions. We assume that we have access to local information that can be summarized via pairwise distributions. Note that our algorithm can be easily extended to arbitrary (non-pairwise) local marginal distributions and we make the assumption for clarity of presentation only. Let $D$ denote the KL divergence (distance) between two probability distributions defined over the same set of variables. Namely, $D(\mathcal{A}, \mathcal{B}) = \sum_{\boldsymbol{x}} \mathcal{A}(\boldsymbol{x}) \log \mathcal{A}(\boldsymbol{x}) -$

$\sum_{\boldsymbol{x}} \mathcal{A}(\boldsymbol{x}) \log \mathcal{B}(\boldsymbol{x})$ where $\mathcal{A}$ and $\mathcal{B}$ are two distributions defined over $\boldsymbol{X}$. The first term equals the negative *entropy* of the distribution $\mathcal{A}$ and the second term equals the negative *cross-entropy* between $\mathcal{A}$ and $\mathcal{B}$.

Let $\mathcal{E}$ denote a subset of pairs of random variables, namely $\mathcal{E} \subseteq \{(X_j, X_k) | X_j, X_k \in \boldsymbol{X} \text{ and } j < k\}$.

**Definition 1.** *We say that a set of local estimates $\{\mathcal{P}_{jk}(X_j, X_k) | (X_j, X_k) \in \mathcal{E}\}$ is **consistent** iff there exists a probability distribution $\mathcal{P}$ such that each estimate $\mathcal{P}_{jk}(X_j, X_k)$ is a marginal probability distribution of $\mathcal{P}$. Otherwise, we say that the set is **inconsistent**.*

Given a set of local estimates and data, the learning problem is given by:

**Given:** A cutset network structure with parameters $\Theta$ representing a probability distribution $\mathcal{R}$, a set of pairwise local distributions $\mathcal{P}_{jk}(X_j, X_k)$ where $(X_j, X_k) \in \mathcal{E}$ and a fully observed dataset $\mathcal{X} = \{\boldsymbol{x}^{(1)}, \ldots, \boldsymbol{x}^{(m)}\}$.

**To do:** Find an assignment of values to all parameters in $\Theta$ such that the negative log-likelihood of $\mathcal{X}$ w.r.t. $\mathcal{R}_\Theta$ is minimized and the KL distance between $\mathcal{P}_{jk}(X_j, X_k)$ and $\mathcal{R}_\Theta(X_j, X_k)$ equals zero for all $(X_j, X_k) \in \mathcal{E}$.

Mathematically, we can express it as:

$$\underset{\Theta}{\operatorname{argmin}} \; - \sum_{\boldsymbol{x}^{(d)} \in \mathcal{X}} \log \mathcal{R}_\Theta(\boldsymbol{x}^{(d)}) \; s.t. \; \forall \; (X_j, X_k) \in \mathcal{E}, \; D(\mathcal{P}_{jk}(X_j, X_k), R_\Theta(X_j, X_k)) = 0 \quad (3)$$

Unfortunately, if the set $\{\mathcal{P}_{jk}(X_j, X_k) | (X_j, X_k) \in \mathcal{E}\}$ is inconsistent (see Def. 1), then the above constrained optimization problem is infeasible.[1] To address this issue, we propose to use a linear combination of the objective and constraints (which is akin to Lagrange relaxation). Mathematically,

$$\underset{\Theta}{\operatorname{argmin}} \; \lambda_1 \sum_{(X_j, X_k) \in \mathcal{E}} D(\mathcal{P}_{jk}(X_j, X_k), R_\Theta(X_j, X_k)) \; - \; \lambda_2 \left( \sum_{\boldsymbol{x}^{(d)} \in \mathcal{X}} \log \mathcal{R}_\Theta(\boldsymbol{x}^{(d)}) \right) \quad (4)$$

where $\lambda_1 \geq 0$ and $\lambda_2 \geq 0$ are hyperparameters (technically, we only need one hyperparameter; we use two for convenience) that model the relative importance of the local and global statistics respectively. Here, $D(\mathcal{P}_{jk}(X_j, X_k), \mathcal{R}_\Theta(X_j, X_k)) = \sum_{x_j, x_k} \mathcal{P}_{jk}(x_j, x_k) \log \mathcal{P}_{jk}(x_j, x_k) - \sum_{x_j, x_k} \mathcal{P}_{jk}(x_j, x_k) \log \mathcal{R}_\Theta(x_j, x_k)$.

### 4.2 Simplifying the Learning Problem

We simplify the optimization problem given in Eq. (4) by making the following observation. Since $\sum_{x_j, x_k} \mathcal{P}_{jk}(x_j, x_k) \log \mathcal{P}_{jk}(x_j, x_k)$ is constant, we can remove it from the objective function and replace each term $D(\mathcal{P}_{jk}(x_j, x_k), R_\Theta(x_j, x_k))$ in Eq. (4) by $-\sum_{x_j, x_k} \mathcal{P}_{jk}(x_j, x_k) \log R_\Theta(x_j, x_k)$.

Negating the objective and making the substitutions described above yields the following maximization problem:

$$\underset{\Theta}{\operatorname{argmax}} \; \lambda_1 \sum_{(X_j, X_k) \in \mathcal{E}} \sum_{x_j, x_k} \mathcal{P}_{jk}(x_j, x_k) \log \mathcal{R}_\Theta(x_j, x_k) + \lambda_2 \left( \sum_{\boldsymbol{x}^{(d)} \in \mathcal{X}} \log \mathcal{R}_\Theta(\boldsymbol{x}^{(d)}) \right) \quad (5)$$

The optimization problem given in Eq. (5) is not concave in the parameters $\Theta$ but it is smooth. Therefore, it can be solved using an iterative, gradient ascent algorithm. However, a drawback of this algorithm is that the gradient of the second term in Eq. (5) requires us to go over the whole data at each iteration yielding a slow algorithm. Although, stochastic gradient descent or mini-batch approaches can be used to address this issue, their convergence is quite slow in practice. Therefore, in order to reduce the computational complexity, we propose the following moment-matching approach.

Let $\mathcal{Q}$ denote the distribution associated with the cutset network having the same structure as $\mathcal{R}$. Thus, there is a one-to-one correspondence between the parameters of $\mathcal{Q}$ and $\mathcal{R}$. Let $\Pi$ denote the set

---

[1]Note that this condition is sound but not complete. For example, the problem is inconsistent if the estimates are consistent but the cutset network does not have enough representation power to faithfully represent the constraints given in Eq. (3).

of parameters of $\mathcal{Q}$. Thus, given a parameter $\pi_{x_i,\boldsymbol{u}_i} \in \Pi$, there is a corresponding parameter $\theta_{x_i,\boldsymbol{u}_i}$ in $\Theta$. Let the set of parameters $\Pi$ be learned from data $\mathcal{X}$ by maximizing the log-likelihood. Since the parameters of cutset networks are conditional probability distributions, given $\mathcal{Q}$ learned from data, we can use negative cross-entropy between parameters of $\mathcal{Q}$ and $\mathcal{R}$ in lieu of the second term (log-likelihood) of Eq. (5). Mathematically,

$$\underset{\Theta}{\operatorname{argmax}} \ \lambda_1 \sum_{(X_j,X_k)\in\mathcal{E}} \sum_{x_j,x_k} \mathcal{P}_{jk}(x_j,x_k)\log\mathcal{R}_\Theta(x_j,x_k) +$$
$$\lambda_2 \sum_{\theta_{x_i,\boldsymbol{u}_i}\in\Theta} \pi_{x_i,\boldsymbol{u}_i}\log\theta_{x_i,\boldsymbol{u}_i} + (1-\pi_{x_i,\boldsymbol{u}_i})\log(1-\theta_{x_i,\boldsymbol{u}_i}) \tag{6}$$

### 4.3 Solving the Learning Problem via Gradient Ascent

We propose to solve the optimization task given in Eq. (6) using gradient ascent methods. To this end, we derive the gradients w.r.t. each parameter $\theta_{x_i,\boldsymbol{u}_i}$. The partial derivative of the second term w.r.t. $\theta_{x_i,\boldsymbol{u}_i}$ is straight-forward and given by

$$\lambda_2 \left( \frac{\pi_{x_i,\boldsymbol{u}_i}}{\theta_{x_i,\boldsymbol{u}_i}} - \frac{1-\pi_{x_i,\boldsymbol{u}_i}}{1-\theta_{x_i,\boldsymbol{u}_i}} \right) \tag{7}$$

The partial derivative of the first term of Eq. (6) is more involved and we summarize it in the following proposition. (Proofs are given in the appendix.)

**Proposition 2.** *The partial derivative of*

$$\lambda_1 \sum_{(X_j,X_k)\in\mathcal{E}} \sum_{x_j,x_k} \mathcal{P}_{jk}(x_j,x_k)\log\mathcal{R}_\Theta(x_j,x_k)$$

*w.r.t. $\theta_{x_i,\boldsymbol{u}_i}$ is given by*

$$\lambda_1 \sum_{(X_j,X_k)\in\mathcal{E}} \sum_{x_j,x_k} \mathcal{P}_{jk}(x_j,x_k) \left( \frac{\mathcal{R}_\Theta(\boldsymbol{u}_i, X_i=1|x_j,x_k)}{\theta_{x_i,\boldsymbol{u}_i}} - \frac{\mathcal{R}_\Theta(\boldsymbol{u}_i, X_i=0|x_j,x_k)}{1-\theta_{x_i,\boldsymbol{u}_i}} \right) \tag{8}$$

## 5 Formal Algorithm

Next, we formally present an algorithm that leverages the gradient equations given in Eqs. (7)–(8) (see Algorithm 1) for solving the learning problem given in Eq. (6). The algorithm, which we call learning cutset networks with local inconsistent statistics (LCN-LIS), takes as input training dataset $\mathcal{X}$, local statistics $P_{jk}(X_j,X_k)$, two hyperparameters $\lambda_1$ and $\lambda_2$ (real numbers) and an integer bound $T$ on the number of iterations. It begins by learning a cutset network $\mathcal{Q}$ from the dataset $\mathcal{X}$ (step 2) and initializes $\mathcal{R}$ to have the same structure as $\mathcal{Q}$ (step 3). In steps 4–19, the algorithm runs the gradient ascent steps. The gradient ascent begins by setting all parameters to a random number between 0 and 1. Then, at each iteration $t$, for each pair $(X_j,X_k)\in\mathcal{E}$ and its possible assignments $(x_j,x_k)$, it sets $X_j=x_j$ and $X_k=x_k$ as evidence and runs a two-pass inference algorithm over the cutset network [5] to compute the required conditional probabilities $\mathcal{R}_{\Theta^t}(\boldsymbol{u}_i, X_i=1|x_j,x_k)$ and $\mathcal{R}_{\Theta^t}(\boldsymbol{u}_i, X_i=0|x_j,x_k)$ (step 9). The algorithm then updates the gradient for each parameter $\theta_{x_i,\boldsymbol{u}_i}$ (steps 10–13; also see Eqs. (7)–(8)) given the assignment $X_j=x_j$ and $X_k=x_k$. In steps 15–17, the algorithm updates the parameters using the gradient estimates $g_{x_i,\boldsymbol{u}_i}$ and learning rate $\alpha$. The algorithm terminates the gradient ascent on convergence or when the bound $T$ on the number of iterations is reached. At termination, the algorithm returns $\mathcal{R}$ with parameters $\Theta^t$.

The main virtue of our algorithm is that it has polynomial computational complexity. The time (and space) complexity of step 9 is $O(|\Theta|)$ using a two-pass algorithm over the CN that calculates the conditional probabilities of the parameters given evidence (see [2, 5]). The time complexity of updating the gradients (steps 10-13) is also $O(|\Theta|)$. Thus, the time complexity of steps 9–13 is $O(|\Theta|)$. Since these steps can be executed a maximum of $O(|\mathcal{E}| \times T)$ times (step 8 and step 6 respectively), where $T$ is the bound on the number of iterations, the overall complexity is $O(|\mathcal{E}| \times T \times |\Theta|)$.

**Remarks.** Note that a feasible solution to the optimization problem given in Eq. (6), and thus the one returned by Algorithm 1 *filters inconsistency* in the local estimates $\{\mathcal{P}_{jk}(X_j,X_k)|(X_j,X_k)\in\mathcal{E}\}$

**Algorithm 1:** LCN-LIS $(\mathcal{X}, \mathcal{E}, \{\mathcal{P}_{jk}(X_j, X_k)|(X_j, X_k) \in \mathcal{E}\}, \lambda_1, \lambda_2, T)$

**Input** : (1) Training examples $\mathcal{X}$ defined over a set of variables $\boldsymbol{X}$; (2) a set of inconsistent pairwise marginal statistics $\mathcal{P}_{jk}(X_j, X_k)$ where $(X_j, X_k) \in \mathcal{E}$ and $\mathcal{E} \subseteq \{(X_j, X_k)|X_j, X_k \in \boldsymbol{X}$ and $j < k\}$; (3) Two hyper parameters $\lambda_1 \geq 0$ and $\lambda_2 \geq 0$, and (4) a bound $T$ on the number of iterations

**Output** : A Cutset Network $\mathcal{R}$

1 **begin**

2    Learn a Cutset Network $\mathcal{Q}$ from Data $\mathcal{X}$ using learning algorithms from literature (cf. [15–17]).

3    Initialize cutset network $\mathcal{R}$ to have the same structure as $\mathcal{Q}$. Let $\Theta = \{\theta_{x_i, \boldsymbol{u_i}}\}$ and $\Pi = \{\pi_{x_i, \boldsymbol{u_i}}\}$ denote the set of parameters of $\mathcal{R}$ and $\mathcal{Q}$ respectively.

4    Initialize: all parameters $\theta_{x_i, \boldsymbol{u_i}}$ of $\mathcal{R}$ to a random number between 0 and 1. Let $\theta^0_{x_i, \boldsymbol{u_i}}$ denote the initial value of $\theta_{x_i, \boldsymbol{u_i}}$

5    Initialize: $t = 0$

6    **repeat**

7      Initialize: $g_{x_i, \boldsymbol{u_i}} = 0$ for each $\theta_{x_i, \boldsymbol{u_i}} \in \Theta$

8      **for** *each possible value assignment* $X_j = x_j$ *and* $X_k = x_k$ *where* $(X_j, X_k) \in \mathcal{E}$ **do**

9        Set $X_j = x_j$ and $X_k = x_k$ as evidence in $\mathcal{R}$ and compute the conditional probabilities $\mathcal{R}_{\Theta^t}(\boldsymbol{u}_i, x_i | x_j, x_k)$ for each $\theta_{x_i, \boldsymbol{u_i}} \in \Theta$

10        **for** *each parameter* $\theta_{x_i, \boldsymbol{u_i}} \in \Theta$ **do**

11          Let:

$$\delta^+ = \frac{\mathcal{R}_{\Theta^t}(\boldsymbol{u}_i, X_i = 1 | x_j, x_k)}{\theta^t_{x_i, \boldsymbol{u_i}}} \text{ and } \delta^- = \frac{\mathcal{R}_{\Theta^t}(\boldsymbol{u}_i, X_i = 0 | x_j, x_k)}{(1 - \theta^t_{x_i, \boldsymbol{u_i}})}$$

12          Update:

$$g_{x_i, \boldsymbol{u_i}} = g_{x_i, \boldsymbol{u_i}} + \lambda_1 \mathcal{P}_{jk}(x_j, x_k)(\delta^+ - \delta^-) + \lambda_2 \left( \frac{\pi_{x_i, \boldsymbol{u_i}}}{\theta^t_{x_i, \boldsymbol{u_i}}} - \frac{1 - \pi_{x_i, \boldsymbol{u_i}}}{1 - \theta^t_{x_i, \boldsymbol{u_i}}} \right)$$

13        **end**

14      **end**

15      **for** *each parameter* $\theta_{x_i, \boldsymbol{u_i}} \in \Theta$ **do**

16        $\theta^{t+1}_{x_i, \boldsymbol{u_i}} = \theta^t_{x_i, \boldsymbol{u_i}} + \alpha \times g_{x_i, \boldsymbol{u_i}}$ // $\alpha$: learning rate

17      **end**

18      $t = t + 1$

19    **until** *convergence or* $t \geq T$;

20    **return** $\mathcal{R}$ *with parameters* $\Theta^t$

21 **end**

because it yields a globally consistent model $\mathcal{R}_\Theta$. Unlike previously proposed techniques for solving the optimization task in Eq. (3) such as the iterative proportional fitting procedure (IPFP) [6, 25] which will not converge when the local estimates are inconsistent [2], Algorithm 1 will converge to a local optimum (because the objective is smooth).

To summarize, we derived and presented a gradient-based algorithm for learning the parameters of cutset networks in presence of local estimates (see Algorithm 1) and showed that the algorithm requires time and space that scales linearly with the number of given local statistics. Note that Theorem 2 can be easily extended to Bayesian and Markov networks. However, the problem is that computing the terms in the numerator in Eq. 8 will be NP-hard in general on Bayesian and Markov networks. This highlights another virtue of tractable models, local information, even if it is inconsistent can be efficiently integrated into a tractable model.

## 6 Experiments

We performed a detailed, controlled experimental study to evaluate the impact of using inconsistent local statistics on the quality of the learned model. Specifically, we used the following controls: (1) the

---

[2]IPFP and its extension called GEMA proposed by Vomlel [25] are computationally infeasible unless the treewidth of the primal graph obtained by combining the edges corresponding to $\mathcal{E}$ and the cutset network $\mathcal{Q}$ is bounded by a small constant and the cutset network is an I-map (cf. [8]) of a Markov network whose potentials are given by $\mathcal{E}$. The second requirement is particularly difficult to satisfy in practice.

accuracy of the cutset network $\mathcal{Q}$ learned from data $\mathcal{X}$ in Algorithm 1; (2) the strength/level of inconsistency in the local statistics $\mathcal{P}_{jk}(X_j, X_k)$; (3) the cutset network architecture; and (4) the number of evidence variables or observations available at test time (to test discriminative performance).

We used 20 benchmark datasets that have been widely used in previous studies [20, 21] to evaluate our new approach. The number of variables in these datasets vary from 16 to 1556, and all variables are binary. We ran Algorithm 1 for a maximum of 48 hours or 1000 iterations (namely $T = 1000$) or convergence,whichever was earlier. We used 5-fold cross-validation to select the values of the hyperparameters $\lambda_1$ and $\lambda_2$.

**Local estimates:** For each dataset, we learned a mixture of cutset networks [16, 17] and used it as the true model $\mathcal{P}$. We generated local statistics from $\mathcal{P}$ as follows. Since $\mathcal{P}$ is a tractable model, we can efficiently (in linear time) compute $\mathcal{P}_{jk}(X_j, X_k)$ for all $X_j, X_k \in \boldsymbol{X}$. To make them inconsistent, we added a value $\epsilon$ that is randomly sampled from a normal distribution with $0$ mean and standard deviation $\sigma$. We experimented with five values of $\sigma : \{0.001, 0.01, 0.05, 0.1, 0.2\}$. Note that after adding $\epsilon$, we have to normalize the distributions to ensure that they are valid.

**Data model:** We used $10\%$ of the randomly chosen examples in the training set to learn $\mathcal{Q}$. Thus, the dataset used by Algorithm 1 has $90\%$ fewer examples than the one used for learning $\mathcal{P}$. We did this to ensure that $\mathcal{Q}$, learned from a much smaller amount of data, differs significantly from the true model $\mathcal{P}$, which in turn will help us evaluate how local information improves the quality from an inferior starting point. We further controlled the quality of $\mathcal{Q}$ using a parameter $h$ which we call the *perturb rate*. $h$ lies

Table 1: Generative (0% evidence) performance measured using the negative cross entropy between $\mathcal{P}$ and $\mathcal{Q}$, and between $\mathcal{P}$ and $\mathcal{R}$ on three models: CLTs, CNs, MCNs.

| Datasets | #var | CLTs | | CNs | | MCNs | |
|---|---|---|---|---|---|---|---|
| | | $\mathcal{Q}$ | $\mathcal{R}$ | $\mathcal{Q}$ | $\mathcal{R}$ | $\mathcal{Q}$ | $\mathcal{R}$ |
| nltcs | 16 | -7.62 | **-6.85** | -6.86 | **-6.17** | -7.32 | **-6.03** |
| msnbc | 17 | -7.01 | **-6.58** | -6.99 | **-6.36** | -7.08 | **-6.08** |
| kdd | 64 | -5.20 | **-2.80** | -5.86 | **-2.77** | -5.52 | **-2.69** |
| plants | 69 | -18.64 | **-17.23** | -17.70 | **-15.55** | -17.04 | **-15.05** |
| audio | 100 | -47.16 | **-45.52** | -47.15 | **-44.01** | -44.20 | **-42.57** |
| jester | 100 | -60.95 | **-59.62** | -61.08 | **-57.45** | -57.92 | **-56.70** |
| netflix | 100 | -62.98 | **-61.46** | -63.85 | **-60.93** | -58.31 | **-57.73** |
| accidents | 111 | -38.00 | **-34.21** | -37.06 | **-33.14** | -35.72 | **-32.43** |
| retail | 135 | -15.54 | **-11.59** | -17.22 | **-11.61** | -16.68 | **-11.61** |
| pumsb* | 163 | -37.23 | **-32.86** | -33.25 | **-27.61** | -38.55 | **-30.46** |
| dna | 180 | -103.18 | **-95.89** | -124.06 | **-99.41** | -98.53 | **-90.58** |
| kosarek | 190 | -21.04 | **-12.98** | -19.46 | **-12.45** | -18.23 | **-10.68** |
| msweb | 294 | -24.57 | **-12.28** | -24.37 | **-12.33** | -22.48 | **-11.05** |
| book | 500 | -58.07 | **-41.80** | -56.28 | **-41.46** | -51.81 | **-34.49** |
| movie | 500 | -97.27 | **-77.21** | -94.39 | **-70.00** | -64.99 | **-34.14** |
| webkb | 839 | -207.12 | **-172.65** | -199.76 | **-162.30** | -181.05 | **-126.06** |
| reuters | 889 | -152.53 | **-113.42** | -146.47 | **-109.21** | -149.49 | **-111.29** |
| 20newsg | 910 | -194.55 | **-157.70** | -181.47 | **-155.48** | -177.83 | **-152.65** |
| bbc | 1058 | -325.51 | **-270.74** | -265.46 | **-210.53** | -243.87 | **-223.60** |
| ad | 1556 | -112.01 | **-47.27** | -93.96 | **-41.34** | -96.79 | **-53.70** |
| **Total AVG** | | -79.81 | **-64.03** | -75.14 | **-59.01** | -69.67 | **-55.43** |

between $0$ and $100$, and given a value for $h$, we replaced $h\%$ of the parameters in $\mathcal{Q}$ with a random number. We normalized $\mathcal{Q}$ to ensure that it is a valid probability distribution.

We used three types of cutset network architectures: (1) cutset networks with depth $0$ which are equivalent to Chow-Liu trees (CLTs); (2) cutset networks with no latent variables (CNs); and (3) mixtures of cutset networks (MCNs). The latter is a state-of-the-art model [19]. We learned both discriminative and generative cutset networks. In discriminative networks, we set $L\%$ of random variables as evidence $\boldsymbol{E}$ and learn a probability distribution over the variables $\boldsymbol{X} \setminus \boldsymbol{E}$ given an assignment $\boldsymbol{e}$ to the evidence variables. We used 4 values for $L : \{0, 20, 50, 80\}$. When $L = 0$, we get a generative model while the remaining models are discriminative.

As our evaluation criteria, we used *negative cross-entropy* between the true $\mathcal{P}$ and the learned model. The higher the negative cross-entropy, the better the model.

## 6.1 Results

**Improved Model Quality.** For lack of space, we present plots showing the impact of the level of noise in $\mathcal{Q}$, controlled by $h$; and the level of inconsistency in the pairwise estimates $\mathcal{P}_{jk}(X_j, X_k)$, controlled by $\sigma$ on one randomly chosen dataset. We present the plots for the remaining datasets in

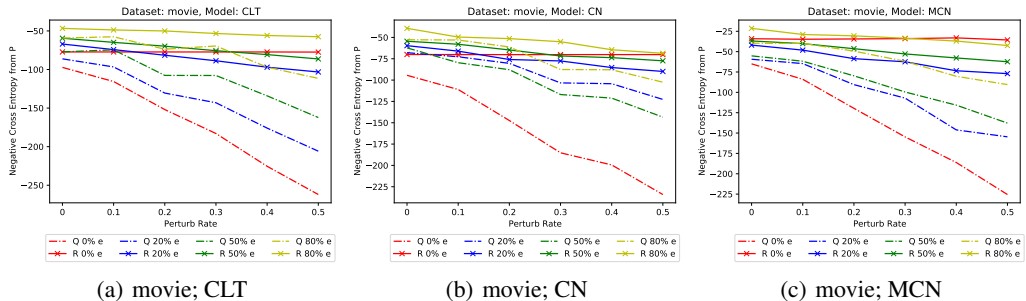

| (a) movie; CLT | (b) movie; CN | (c) movie; MCN |

Figure 2: Negative Cross Entropy between $\mathcal{P}$ and $\mathcal{Q}$, and between $\mathcal{P}$ and $\mathcal{R}$ with evidence of 0%, 20%, 50%, and 80% on three different models: CLT, CN, and MCN, as a function of perturb rate for a randomly chosen dataset: movie.

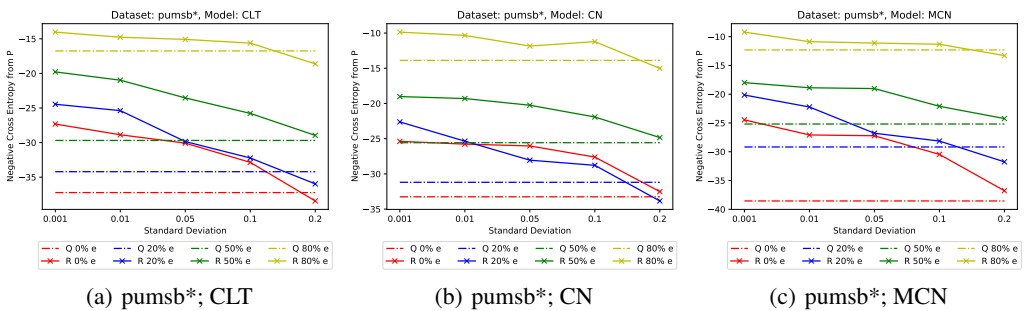

| (a) pumsb*; CLT | (b) pumsb*; CN | (c) pumsb*; MCN |

Figure 3: Negative Cross Entropy between $\mathcal{P}$ and $\mathcal{Q}$, and between $\mathcal{P}$ and $\mathcal{R}$, with evidence of 0%, 20%, 50%, and 80% on three different models: CLT, CN, and MCN, as a function of standard deviation $\sigma$ (of Gaussian noise that is applied to the local statistics $\mathcal{P}_{jk}(X_j, X_k)$) on one randomly chosen dataset: pumsb*.

the supplementary material. Figure 2 presents the negative cross-entropy scores achieved by CLTs, CNs, and MCNs as a function of the perturb rate ($h$) for a randomly chosen dataset 'movie', and for different values of $L$ (the percentage of evidence variables). These plots help us evaluate the impact that the quality of $\mathcal{Q}$ has (namely the impact of the starting point) on the model learned by Algorithm 1. We observe that as the perturb rate increases, there is a substantial drop in the negative cross-entropy between $\mathcal{P}$ and $\mathcal{Q}$. However, the negative cross entropy between $\mathcal{P}$ and $\mathcal{R}$ remains relatively flat. This shows that the use of local statistics significantly improves the model quality, especially when the model based on global data alone is inaccurate.

Figure 3 shows the negative cross-entropy scores achieved by CLTs, CNs and MCNs as a function of the standard deviation $\sigma \in \{0.001, 0.01, 0.05, 0.1, 0.2\}$ of the Gaussian noise that is applied to the local statistics $\mathcal{P}_{jk}(X_j, X_k)$ for the dataset 'pumsb*', and for different values of $L$ (the percentage of evidence variables). The figure clearly shows that when the local statistics contain more noise, the performance degrades as expected. When the $\sigma$ is large ($\sigma > 0.2$), there is a non-zero chance that the final model $\mathcal{R}$ will be worse than $\mathcal{Q}$. Thus, based on our experimental results, our proposed algorithm is more likely to yield a better model than the one based on global information alone when the Gaussian noise has standard deviation $\leq 0.1$.

**Improved Generative and Discriminative Performances.** Tables 1 and 2 show the generative and discriminative ($L = 50$ and $L = 80$) performances of $\mathcal{Q}$ and $\mathcal{R}$ respectively. We present the results for $L = 20$ in the supplement. We used $\sigma = 0.1$ and $h = 0$ to generate the experimental data given in the tables. Each value in the tables is an average over 5 runs. To avoid clutter, we do not report the standard deviation because it was fairly small over all the runs. These results help us analyze the impact of the cutset network architecture and the number of evidence variables on our evaluation criteria. We observe that, on average, using local inconsistent statistics improves the negative cross entropy of each architecture by 17-23%. MCNs are the best performing model overall

Table 2: Discriminative (50% and 80% evidence) performance of $\mathcal{Q}$ and $\mathcal{R}$ as measured using negative cross-entropy on three models: CLTs, CNs, MCNs.

| Dataset | 50% Evidence | | | | | | 80% Evidence | | | | | |
|---|---|---|---|---|---|---|---|---|---|---|---|---|
| | CLTs | | CNs | | MCNs | | CLTs | | CNs | | MCNs | |
| | $\mathcal{Q}$ | $\mathcal{R}$ | $\mathcal{Q}$ | $\mathcal{R}$ | $\mathcal{Q}$ | $\mathcal{R}$ | $\mathcal{Q}$ | $\mathcal{R}$ | $\mathcal{Q}$ | $\mathcal{R}$ | $\mathcal{Q}$ | $\mathcal{R}$ |
| nltcs | -4.14 | **-3.53** | -4.03 | **-3.86** | -3.53 | **-3.18** | -0.45 | **-0.27** | -0.42 | **-0.33** | -0.44 | **-0.29** |
| msnbc | -5.14 | **-4.68** | -5.34 | **-5.30** | -5.03 | **-2.14** | -4.02 | **-2.77** | -3.24 | **-2.84** | -3.36 | **-2.70** |
| kdd | -4.28 | **-2.97** | -5.12 | **-2.89** | -3.82 | **-2.19** | -3.51 | **-1.94** | -3.02 | **-1.67** | -2.99 | **-1.68** |
| plants | -16.07 | **-8.75** | -16.32 | **-9.72** | -15.15 | **-8.28** | -10.27 | **-7.20** | -10.23 | **-7.70** | -9.35 | **-6.64** |
| audio | -32.14 | **-26.68** | -31.59 | **-30.76** | -28.83 | **-25.93** | -27.23 | **-25.88** | -21.55 | **-17.76** | -21.02 | **-18.20** |
| jester | -49.55 | **-45.73** | -49.1 | **-48.08** | -44.55 | **-42.27** | -44.68 | **-29.78** | -35.07 | **-34.24** | -35.51 | **-27.13** |
| netflix | -56.43 | **-56.14** | -51.97 | **-48.84** | -48.28 | **-46.68** | -55.29 | **-53.86** | -47.55 | **-46.98** | -46.38 | **-45.28** |
| accidents | -34.27 | **-31.73** | -33.03 | **-29.3** | -30.1 | **-28.34** | -34.19 | **-28.96** | -32.36 | **-27.84** | -29.53 | **-25.56** |
| retail | -11.75 | **-9.02** | -12.44 | **-11.27** | -11.07 | **-9.32** | -8.86 | **-6.08** | -8.09 | **-5.95** | -7.38 | **-4.98** |
| pumsb* | -29.69 | **-25.78** | -25.57 | **-21.92** | -25.18 | **-22.10** | -16.75 | **-15.62** | -13.88 | **-11.22** | -12.31 | **-11.32** |
| dna | -90.48 | **-82.46** | -107.81 | **-88.33** | -85.73 | **-74.24** | -79.33 | **-65.40** | -99.28 | **-82.15** | -75.45 | **-60.68** |
| kosarek | -12.62 | **-9.00** | -11.95 | **-5.73** | -9.23 | **-4.17** | -10.38 | **-8.63** | -8.82 | **-4.40** | -7.87 | **-5.22** |
| msweb | -12.28 | **-8.18** | -11.97 | **-9.89** | -10.27 | **-9.35** | -11.66 | **-5.66** | -7.72 | **-6.87** | -7.72 | **-6.31** |
| book | -23.42 | **-15.05** | -20.42 | **-18.05** | -20.49 | **-13.96** | -17.36 | **-15.17** | -15.83 | **-13.05** | -14.54 | **-11.72** |
| movie | -77.15 | **-59.46** | -62.22 | **-54.47** | -55.2 | **-36.89** | -59.2 | **-46.65** | -52.7 | **-39.36** | -39.65 | **-21.40** |
| webkb | -167.71 | **-152.22** | -155.01 | **-139.86** | -134.67 | **-123.38** | -105.83 | **-72.08** | -91.48 | **-61.68** | -90.19 | **-55.65** |
| reuters | -113.17 | **-91.79** | -105.21 | **-85.56** | -110.26 | **-89.12** | -83.56 | **-68.86** | -74.57 | **-62.59** | -77.75 | **-65.24** |
| 20newsg | -157.03 | **-128.89** | -143.25 | **-139.89** | -144.61 | **-121.72** | -93.59 | **-85.91** | -87.6 | **-74.44** | -79.68 | **-48.44** |
| bbc | -247.79 | **-209.23** | -243.75 | **-174.1** | -241.93 | **-184.89** | -171.31 | **-163.04** | -164.16 | **-148.02** | -167.02 | **-158.23** |
| ad | -88.33 | **-34.38** | -65.31 | **-34.53** | -72.13 | **-34.78** | -80.7 | **-26.19** | -57.95 | **-30.09** | -55.6 | **-27.13** |
| Avg | -61.67 | **-50.28** | -58.07 | **-48.12** | -55 | **-44.15** | -45.91 | **-36.5** | -41.78 | **-33.56** | -39.19 | **-30.19** |

and Chow-Liu trees (CLTs) are significantly worse. There was no significant difference in the amount of improvement as we increased the number of evidence nodes. This suggests that our approach is equally useful for both discriminative and generative models.

## 7 Conclusion

In this paper, we presented a new method for learning the parameters of cutset networks in presence of inconsistent local estimates. Unlike conventional algorithms which use full i.i.d. data during the learning process, we proposed a novel approach that uses noisy local information to learn a more accurate and robust model. The key advantage of using local estimates is that they are often readily available as compared to full i.i.d. data. We also showed via experiments on benchmark datasets that our new algorithm greatly improves the quality of the initial model learned from i.i.d. data, even when the local estimates are inconsistent and noisy.

Future work includes: (1) developing new structure learning algorithms that use local inconsistent statistics; (2) applying our approach to other tractable models such as sum-product networks and probabilistic sentential decision diagrams; (3) improving human feedback to improve explanation quality; (4) developing lazy learning algorithms; etc.

## Acknowledgments and Disclosure of Funding

This work was supported in part by the DARPA Explainable Artificial Intelligence (XAI) Program under contract number N66001-17-2-4032, the DARPA Perceptually-Enabled Task Guidance (PTG) Program under contract number HR00112220005 and by the National Science Foundation grants IIS-1652835 and IIS-1528037.

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
