# OpenReview forum: "Learning Tractable Probabilistic Models from Inconsistent Local Estimates"
_NeurIPS.cc/2022/Conference — NeurIPS 2022 Accept_

### Official Review · Reviewer_gNZo · 2022-06-25

**Rating:** 3
**Confidence:** 5
**Soundness:** 3 good
**Presentation:** 3 good
**Contribution:** 1 poor

**Summary:**

This paper proposes to incorporate so-called local estimates into the optimization of probabilistic models. Since such estimates are typically intractable, the paper focuses on using tractable probabilistic models in order to enforce them into the loss function. The main discussion centers on learning cutset networks with (possible inconsistent) local estimates. To do so the paper formulates this as a constrained optimization problem (constraints being the local estimates) and takes a Lagrangian relaxation approach which basically ends up looking like a weighted joint optimization problem. Finally, due to the properties of cutset networks where parameters map to conditional probabilities, the loss function variance can be reduced through moment-matching instead of SGD, leading to faster convergence. Experiments are run on the 20-Datasets benchmark, showing that incorporating local estimates leads to better learned models when evaluated on KL divergence against an oracle model (chosen to be a mixture of cutset networks).

**Questions:**

.

**Limitations:**

.

**Strengths And Weaknesses:**

My overall opinion of this paper is poor. The paper has three significant and pervasive shortcomings.

From start to finish, the assumption of local estimates felt impractical and not well-justified. Are there really many scenarios where we would have large access to local estimates? If so, why are there no experiments on real datasets with real local estimates? The current experiments are completely synthetic, in the sense that the paper learns an oracle model using the full dataset to synthetically generate the local estimates, and then synthetically hides partial dataset from the learner models.

Why are the experiments only testing Q vs R, with no reasonable baselines? It is completely expected that R will do better than Q, since it is given strictly more information (essentially local statistics from the oracle model)? Where are the ablations using SGD, using other TPMs, or using any other method to incorporate local estimates? Altogether, I find it the experimental evaluation sorely lacking.

Why is there such emphasis on cutset networks? The paper tries to start off general, making statements that try to encompass all TPMs, but the entirety of the experiments end up being focused on cutset networks. Even the moment-matching technique is specific to cutset networks. General TPMs do not have the conditional distribution interpretation in their parameters -- would probably need to rely on SGD for optimization. Given this, there should absolutely have been studies ran on other families of TPMs. As it stands, the paper (and its title) should advertise only cutset networks.

- on this, why don't we use general probabilistic circuits? What are the drawbacks of PCs compared to cutset networks?



**Strengths**

- the formulation is reasonable, and the derivations look correct
- the writing is reasonably clear

**Weaknesses**

- poor justification of access to local estimates. At the very least there should be experiments on datasets with real local estimates, not synthetically generated.
- experiments section is weak and far from 'detailed'. Missing ablations (SGD instead of moment matching), other TPMs, choices of oracle model...
- unjustified choice of the heavy emphasis on cutset networks.

---

> ### Author Response · Authors · 2022-08-02
> **We have tried to address your concerns**
>
> Thank you for your review. I guess you were extremely disappointed with the paper. Hopefully, the following helps alleviate your concerns:
>
> 1. We are repeating the answer given to other reviewers. The setting is quite realistic. We have described three applications in the supplement (see section C). Also, various papers we have cited in the related works section use this setting. A fourth application where this setting is useful is "explainable AI models." For example, when the model provides an unsatisfactory explanation, the user may correct the explanation by providing a new estimate. Having said this, we would like to acknowledge that most real-world data will be proprietary where you have local interaction among variables (users of a social net for instance) and the model is trying to build a global picture (a joint distribution).
>
> 2. In fact, we don't know that R will be better than Q since we are only given local information (We can construct cases where local estimates will actually hurt if the model is not an I-map of P and/or large amount of data is used). We considered all ablations you have mentioned: (a) SGD, (b) using other TPMs, or (c) using any other method to incorporate local estimates. Details below:
>   (a) SGD is very unstable when combined with local estimates. We had to use large mini-batch sizes which defeats the purpose of using SGD. We can speculate why (informally). At each iteration of (mini-batch) SGD, we would make a move that balances the current estimate from the mini-batch with the local estimates. When the two differ significantly, there will be large jump/fluctuation.  Therefore, we didn't compare with an unstable, high complexity procedure.
> (b) Well, we did not compare with other TPMs because of a technical problem. The current code/software for these TPMs will make our method a factor of O(n) slower. We detail this technical problem in point #3 (Why cutset networks?)
> (c) See our note about related work in the paper. We would like to point out that existing methods are impractical when the treewidth of the local estimate graph is large (see the related work section and footnotes 2 and 3 in the paper). Also, see our explanation for the first reviewer (GEMA/IPF). Therefore, we did not compare except on the smallest dataset (nltcs).
>
> 3. So here is the technical issue. Consider a simple task. Let us say I give you the simplest tractable model, a tree Markov or Bayesian network and your task is to compute all pairwise marginals over the network (this is required for computing the gradients in our algorithm). If you use the junction tree algorithm, you can solve this problem in $O(n^2d^3)$ time where $n$ is the number of features/variables and $d$ is the domain size of each variable (this is an exercise in Koller and Friedman book, 2009). However, if you use a variable elimination algorithm, this complexity is $O(n^3d^2)$ (the complexity of computing each estimate is $O(nd^2)$ and since there are $n^2$ estimates, the overall complexity is $O(n^3d^2)$). Unfortunately, existing PCs and more importantly software for them use the variable elimination style approach for answering queries, which unnecessarily adds a factor of $O(n)$. Cutset networks are more amenable to using the junction tree style procedure (see the code in the supplement where we use a database/collection of junction trees for fast processing).
> To make a long story short, why not other PCs? Because we can only apply our technique to small datasets unless the current software for PCs is updated. This can be done, just requires significant effort (in other words, our method still applies but we need to change existing PC software). There is a novel problem for PCs we have to solve: given a collection of local marginals that you need to compute and a PC, define a fast computation graph for finding the marginals such that the complexity is much smaller than $O(C \times PCSize)$ where $C$ is the size of the collection and $PCSize$ is the size of the PC.
>
> Finally, as far as we know, cutset networks are SOTA or close to SOTA on all the datasets used in the paper.

---

> > ### Comment · Reviewer_gNZo · 2022-08-03
> > **Response to response**
> >
> > Thank you for the response. I have read it, and the other reviews/responses.
> >
> > 1. I am not totally convinced by some of the listed settings where you claim that we do have access to local estimates. But this alone may be fine since there are definitely applications that I am not too familiar with. However, what I still find to be sorely lacking is that the experimental setup is very synthetic with the use of oracle model to generate estimates, and hiding data from learner models. This seems to be a long throw away from the real setup that the paper aspires to address. If the authors claim that there are so many potential realistic applications, it seems unacceptable that the paper does not evaluate on even a single one.
> >
> > 2. I still have a hard time appreciating the experiments done in this paper, with essentially no baselines and just showing that R does better than Q. I also think that the statement "we can construct cases where local estimates will actually hurt if the model" tells us very little, especially as I imagine these constructions are rarely relevant to real settings. In any case, if you really want the main message of the experiments to be that R does better than Q (which I find not super interesting), then it seems almost imperative that you motivate these cases where local estimates can hurt, and address why we should care about these cases (how relevant, how frequently do they occur in real applications).
> >
> > 3. What is the size of $C$ in your experiments? I would think that $O(C \times PCSize)$ is quite practical to run, and I think PCs are very suitable for this problem formulation. Based on my understanding, this should be fairly scalable and easy to try out.

---

> > > ### Author Response · Authors · 2022-08-03
> > > **Let us try to explain further**
> > >
> > > **Response to #1:** Let us be more specific on two applications: *lazy learning of conditional tractable models* and *explanation feedback*
> > > + In lazy learning of conditional cutset networks or conditional SPNs, we learn $P(Y|X=x)$ from local classifiers. Consider a case where you learn local classifiers $P(Y_i,Y_j|X)$ for all pairs $Y_i, Y_j \in Y$ (here $X$ are the features and $Y_i,Y_j$ are the class variables). These local statistics are typically accurate because classifiers are good at predicting a few variables given others but have a hard time predicting a joint distribution over a large number of variables (think multi-class with $4$ classes versus $2^k$ classes such that $k>2$) The question is can these local statistics help you learn a better model given $X$. We have considered this setting in the experiments where noisy perturbation of $(Y_i,Y_j)$ is available. Here, $Q$ which models $P(Y|X)$ is learned from data and $R$ is updated using these local estimates
> > > + In explanation feedback, $Q$ is the model learned from data, $P$ is the mental model about the world that the user has, but can only provide local estimates and $R$ is the model that updates $Q$ given local estimates from the user's mental model. Consider a simple application where the input is a surveillance video and your task is to explain to a user why you detected a particular activity as suspicious. The explanation according to $Q$ (learned from data) might be the Person touched an object and looked around for cameras. The user can correct this explanation by saying that the video does not show the person looking around for cameras (the local estimate of the marginal probability of person looking around given evidence). In order to get as close as possible to the user's mental model, you have to update $Q$ based on the local estimate. We are collecting data on a similar application (we have morphed the application to maintain anonymity); its pending IRB review.
> > > -----
> > >
> > > **Response to #2 (No baselines)**
> > > We have mentioned several times in the paper the problem with existing work on local estimates. None of the schemes are practical because of high computational complexity and can only be used on the nltcs dataset in our setup.
> > > + Let us consider the GEMA algorithm, it is exponential in the treewidth of the graph given by the local estimates. In lazy learning application described above, the treewidth of the graph is $n-1$.
> > > + Let us consider your argument: it is not interesting that "R" is better than "Q" because you have more information. But the question still remains, how will you use the local information? Which method will you use? Existing algorithms are impractical in this setting when the amount of local information is large and what we are proposing is a practical method for tractable models.  (Our method is intractable for arbitrary Bayes nets and Markov nets but that is for future work).
> > >
> > > -----
> > >
> > > **Response to #3 (Size of $C$):**
> > > $C$ can be as high as $O(n^2)$. Assuming that the PC size is around $O(n^2)$ (this is generous lower bound on the size of the PC), the complexity of one iteration of the gradient method is $O(n^4)$. Assuming 1000 variables and storing parameters using floats (64 bits), the size of the computation graph will be 8000 GB ($8 \times 10^{12}$). Again, this is a lower bound, python and other languages will have some more overhead. This is for one iteration of gradient descent. We do not know how many iterations we will need in advance. We are not saying that PCs will be impractical; just that we will need to update the software for answering such queries efficiently.
> > >
> > > -----
> > > Finally, thank you for asking questions; this is indeed useful. The summary is that this is a new setup inspired by applications. Prior work ignored computational complexity aspects and experimented with models having 10-50 variables and very small number of local estimates (see the cited work). Thus, there are no real baselines available. The experiments are done under two constraints: noisy local estimates and lazy learning (with evidence experiments). Only cutset networks are used because existing software on PCs is "variable elimination based rather than compilation or junction tree based."

---

> > > > ### Comment · Reviewer_gNZo · 2022-08-07
> > > > **Response to clarifications**
> > > >
> > > > Thank you for the response, which I've read.
> > > >
> > > > After some thought, I think I maintain my original evaluation of the paper. I appreciate the detail in your response, but I think the core of my concerns have not been addressed. Indeed, potential applications are being suggested and possible shortcomings of competing methods are being hypothesized in these responses, but in my opinion none of them have actually been evaluated on in the paper itself in any substantial / scientific way.
> > > >
> > > > Experiments on at least one of these applications and comparison to at least one reasonable baseline (even if they have the shortcomings as you suggest) are absolutely necessary for this paper.

---

> > > > > ### Author Response · Authors · 2022-08-07
> > > > > **No problem: Thoughtful evaluations are indeed useful**
> > > > >
> > > > > Our main goal in our response was to help you in your evaluation. It is perfectly fine (and we really mean it) if you don't want to change your evaluation; we can agree to disagree.
> > > > >
> > > > > We want to mention that we are not hypothesizing shortcomings of existing methods as you have suggested above. We have provided simple formal arguments in the paper based on I-MAP properties and computational complexity which show that existing schemes are impractical (we don't have to explicitly state them as theorems because they are very easy to derive). Based on these arguments, for the lazy learning application and the XAI mental models application, you cannot use existing methods based on computational complexity alone.
> > > > >
> > > > > We are currently working on an approximation of IPFP to get around this computational complexity issue. However, there are several problems that arise when we make approximations: no convergence guarantees; the method is highly dependent on ordering of the constraints; etc. This will be a separate paper.

---

### Official Review · Reviewer_WY8E · 2022-07-09

**Rating:** 7
**Confidence:** 4
**Soundness:** 4 excellent
**Presentation:** 4 excellent
**Contribution:** 3 good

**Summary:**

The paper proposes an algorithm for learning a tractable probabilistic model (TPM) from a dataset consisting of (1) a (possibly small) complete dataset and (2) (possibly many) local marginal estimates.  The tractable probabilistic model is a cutset network (or mixtures of).  They propose a tractable gradient algorithm for learning such a model, and show empirically how their algorithm can significantly improve the quality of a learned model when local marginal estimates can be taken into account.

Cutset networks are tractable; given complete data their parameters can be learned in closed form, and otherwise their gradients can also be computed in polynomial time.  The learning algorithm is based on gradient descent, where the local marginal estimates are incorporated as a soft constraint in the objective function.

As the authors have mentioned, the learning problem (or variations of it) have been considered before in the probabilistic graphical models literature, but there are some fundamental difficulties with this learning task on Bayesian networks and Markov networks.  Namely, checking whether local marginal constraints are satisfied requires inference in the network, which is computationall hard for BNs/MNs but can be done in polytime on tractable models such as cutset networks.

To my knowledge, the exploitation of a tractable probablistic model (TPM) for this learning task is novel.  There are certainly practitioners who are interested in this learning task for their applications, and I believe this type of paper would help spread the practical use of TPMs.


**Questions:**

I am familiar with this setting, and the developments to me were clear enough to me with no big surprises.


**Limitations:**

no negative societal impact, and otherwise limitations and relations to related work were satisfactory.

**Strengths And Weaknesses:**

The proposed problem (learning a probabilistic model from data + marginals) is relevant in practice, and the authors apply a more modern class of (tractable) statistical models that is able to overcome limitations of classical models (like Bayesian networks).  The experiments demonstrate the utility on the learning problem.  Because of the (computational) limitations inherent in Bayesian networks, one did not see this learning scenario as often in practice, but given the use of tractable models, it may see increased visibility.

---

> ### Author Response · Authors · 2022-08-02
> **A brief comment on your review**
>
> In a way, you have precisely stated the main reason for writing this paper. The problem setting has been explored for graphical models, but has high computational complexity. We have extended it to tractable models and showed that for tractable models, we can control the computational complexity and still obtain a fairly stable/good performing learning algorithm.

---

### Official Review · Reviewer_QR7n · 2022-07-11

**Rating:** 7
**Confidence:** 4
**Soundness:** 3 good
**Presentation:** 4 excellent
**Contribution:** 3 good

**Summary:**

This paper investigates a learning setting where training data is paired with noisy local estimates, focusing on Cutset Networks, a popular class of tractable probabilistic models (although the results seem easily extendable to other TPM classes).

Tractable models enable the efficient evaluation of the gradients required to solve this learning problem. The paper then proposes an iterative, gradient ascent approach. To further speed up convergence, instead of maximizing the original objective involving both the local estimates and training data, the paper proposes a moment-matching variant where the local estimates are considered as a post-processing step after training the CN on data using standard techniques.


**Questions:**

1- Can you further substantiate that local estimates can be easily obtained in many real-world applications?

2- Why not including a comparison of the proposed moment-matching approach with a variant where stochastic gradient ascent is used to optimize the objective in Eq. 5?

---

Minor comment:

"To avoid clutter, we do not report the standard deviation because it was fairly small over all runs."

- I would still report how (relatively) small it is or add the full table in the supplementary material.



**Limitations:**

I did not find any unaddressed limitations in the paper. I am looking forward to reading an answer to Q1 in order to better assess the impact of this work, possibly raising my 'Contribution' and overall rating.

**Strengths And Weaknesses:**


I found the paper generally well-written. I think it is quite accessible even for non-experts in tractable models while being sufficiently detailed in the description of the contribution. The empirical evaluation is also convincing.

---

I am not completely convinced that having access to these local estimates is "quite common in real-world applications". The work cited
at lines 38-39 is not specifically supporting the claim. Considering that prior work in this setting is quite limited (according to the Related Work section), it would be nice if the paper substantiated more this claim with real-world examples.

[Updated after rebuttal] The authors addressed my concerns in their response and included the changes in an updated version of the paper.

---

> ### Author Response · Authors · 2022-08-02
> **Answers to your questions included**
>
> Thank you for your review. We have tried to answer your questions below. Please let us know if things are unclear.
>
> 1. We are repeating the answer given to other reviewers. The setting is quite realistic. We have described three applications in the supplement (see section C). Also, various papers we have cited in the related works section use this setting. A fourth application where this setting is useful is "explainable AI models." For example, when the model provides an unsatisfactory explanation, the user may correct the explanation by providing a new estimate. Having said this, we would like to acknowledge that most real-world data will be proprietary where you have local interaction among variables (users of a social net for instance) and the model is trying to build a global picture (a joint distribution).
>
> 2. Stochastic gradient descent was very unstable and usually we had to go with a very large batch size (e.g., in 1000s) in order to obtain a stable procedure. However, for such large batch sizes, there are virtually no complexity savings. Our moment-matching approach on the other hand goes over the tractable model just once and therefore is quite fast.

---

> > ### Comment · Reviewer_QR7n · 2022-08-07
> > **Response**
> >
> > Thank you for your answer. I read the three scenarios included in the supplementary material, they indeed clarify the setting, in my opinion.
> > I understand that you need to focus on the technical aspects (footnote at p.2), but I would still try to include (or hint at) a motivating application in the main text if you find space.
> > I think that your answer and changes to the text addressed my main concern with this work, I will adjust my score accordingly.

---

> > > ### Author Response · Authors · 2022-08-07
> > > **Thank you for the update**
> > >
> > > Thank you for updating your review. We will have an extra page if the paper is accepted. Thank you for the suggestion: we will include motivating applications in more detail and also why existing methods will be impractical. We will also include a detailed discussion in the supplement about these issues.

---

### Official Review · Reviewer_L5Sp · 2022-07-11

**Rating:** 7
**Confidence:** 4
**Soundness:** 3 good
**Presentation:** 3 good
**Contribution:** 3 good

**Summary:**

The paper considers the problem of learning tractable probabilistic models (TPMs) and presents a gradient-based method for learning a TPM from local (possibly inconsistent) estimates. Specifically, a TPM is first learnt from the input dataset and is subsequently refined by updating its parameters using a set of local probability estimates defined over small subsets of variables (i.e., local marginal distributions). The parameters are updated using a gradient descent based approach with an objective function that ensures closeness to the local estimates as well as the learnt model. The empirical evaluation demonstrates that the proposed method produces tractable models that are superior to those learned solely from data.


**Questions:**

1) How realistic is the assumption that the marginal probabilistic estimates are available? Could you describe a real-world application where this information is readily available? The experimental setup in Section 5, especially the generation of the local estimates, looks a bit artificial.

2) I was wondering how previous methods like IPFP perform at least for the case when the local estimates are consistent.

3) How sensitive is the method to the learning rate? Did you try using an adaptive learning rate?

[Post Rebuttal] Thanks for your answers. They addressed my concerns.

**Ethics Review Area:**

["I don’t know"]

**Limitations:**

see above

**Strengths And Weaknesses:**

The paper is fairly well written and organised. The quality of the presentation is overall very good and therefore the paper is relatively easy to follow. Most of the concepts are introduced and discussed if a fairly clear manner. The empirical evaluation is sound and demonstrates clearly the strength of the proposed method.

---

> ### Author Response · Authors · 2022-08-02
> **Addressed the three questions you have**
>
> First of all, thank you for the review. We have tried our best to address your questions. Please let us know if things are not clear.
>
> 1. In fact, the setting is quite realistic. We have described three applications in the supplement (see section C). Also, various papers we have cited in the related works section use this setting. A fourth application where this setting is useful is "explainable AI models." For example, when the model provides an unsatisfactory explanation, the user may correct the explanation by providing a new estimate. Having said this, we would like to acknowledge that most real-world data in the domain will be proprietary.
> 2. Actually, the complexity of IPFP is very high even on tractable models; it is exponential in the number of variables. Thus, even if the estimates are consistent, the method is impractical. Vomlel proposed an alternative idea, an approximation called GEMA which is exponential in the treewidth of the "local estimate graph." Thus, if a large number of local estimates are available, even GEMA is impractical. (note the difference here, the model is tractable, but the local estimate graph may be intractable because the latter has high treewidth).
> We have experimented with GEMA and IPF methods on the nltcs dataset (where they are feasible) and the results are comparable with the method proposed in the paper. We will add that result in the supplement.
> 3. We did not try using an adaptive learning rate or a more fancy optimizer (we will try this if the paper is accepted, it is one line change in the code). If you see the code in the supplement, we used a standard optimizer in sckit-learn/scipy to obtain the results.

---

### Meta-Review · Area_Chair_8hEM · 2022-08-23

**Recommendation:** Accept
**Confidence:** Less certain

**Metareview:**

This submission did not reach a full agreement among PC members. I will not repeat the arguments here as they can be read in the reviews (please see e.g. review WY8E, which according to the authors captures well their intention). The main open criticism is the lack of a real application where the type of assumption used in the paper is present (the other important one being the comparison with other approaches, which seems to have been justified by the authors to a good extent). The concern was originally about the existence of those applications, but later this was resolved (it remained as a concern that the real application is not shown in this work). I consider that this is small when weighted against the positive comments in all reviews.

**Award:**

No

---

### Decision · Program_Chairs · 2022-09-14

Accept